# Determinants of Behavioral Intentions in the Context of Sport Tourism with the Aim of Sustaining Sporting Destinations

**Yunduk Jeong [1], Suk-Kyu Kim [2]**  **and Jae-Gu Yu [3],***

[1]  Department of Sport Management, Kyonggi University, Suwon 16277, Korea; fcgangwon@nate.com
[2]  Department of Sports Science, Dongguk University, Gyeongju 38066, Korea; skkim2018@dongguk.ac.kr
[3]  Department of Sport Industry, Chungang University, Anseong 17546, Korea
*   Correspondence: unlisted@cau.ac.kr

**Abstract:** This study was undertaken to examine structural relationships between event quality, tourist satisfaction, place attachment, and behavioral intentions with emphasis on the mediating effects of tourist satisfaction and place attachment on relations between event quality and behavioral intentions in the context of a small-scale recurring sporting event. Responses obtained from 350 attendees were collected and analyzed. Results showed positive impacts of (a) event quality, tourist satisfaction and place attachment on behavioral intentions, (b) event quality and tourist satisfaction on place attachment, and (c) event quality on tourist satisfaction, and demonstrated (d) tourist satisfaction and place attachment partially mediate relationships between event quality and behavioral intentions and that (e) place attachment partially mediates the relationship between tourist satisfaction and behavioral intentions.

**Keywords:** event quality; tourist satisfaction; place attachment; behavioral intentions; sport tourism; sporting event

## 1. Introduction

Intense competition between countries and regions in the tourism industry has resulted in focus on behavioral intentions, such as intention to recommend and revisit, being viewed as a prime concern of tourism destinations and their marketing organizations [1,2]. In particular, since the number of people using the Internet has dramatically increased, intention to recommend has become increasingly important as tourists can easily disseminate frank opinions, experiences, and knowledge of visited destinations worldwide through social media using, for example, personal blogs, Facebook, YouTube, and Instagram [3]. Thus, it is critical that destination managers consider behavioral intentions a priority for future success.

Destination marketing organizations should attempt to develop effective and attractive tourism marketing strategies to arouse potential tourists' interests and persuade tourists to visit to a destination [4]. The majority of scholars suggest hosting sporting events provides tourists, especially sports fans, an array of enjoyable and memorable experiences [5]. Sporting events are referred to as hedonic services that offer many tangible and intangible benefits to host countries and regions [6]. Notably, it has been reported small-scale sporting events can attract many domestic tourists and improve destination image almost as effectively as mega sporting events [5]. For this reason, small local communities not able to host large-scale sporting events should consider small-scale recurring sporting events a cost-effective means of attracting tourists.

Given that behavioral intentions are important for sustainable destination management in the context of tourism, destination managers should understand how tourists' behavioral intentions are

formed. Increasing numbers of researchers have concluded tourist satisfaction is probably a key factor of behavioral intentions [7], and it is widely acknowledged that customer satisfaction is crucial for assessing the success of marketing strategies. For this reason, research during the past two decades has focused on this concept and explored its antecedents and consequences [8]. Additionally, the link between place attachment and behavioral intentions is viewed as being of considerable importance in tourism marketing literature [9]. The notion of place attachment has received increasing attention from destination marketers over recent years because the concept represents emotional bonds between tourists and places they have visited and is regarded as a definitive tourism marketing strength [10] that might induce behavioral intentions like loyalty.

Although a number of studies have provided a necessary foundation for sport tourism studies, they have three major limitations. First, few published studies have identified positive relationships between event quality and destination-related variables (e.g., place attachment) or outcome variables (e.g., behavioral intentions). In other words, despite the importance of event quality in the context of attracting sport tourists [11], scant empirical work has been conducted to address whether quality has direct consequences. Thus, the present study sought to expand current theorizations by examining the mutual relationships between event quality, tourist satisfaction, place attachment, and behavioral intentions in response to recent requests from sport tourism researchers to provide more exhaustive analyses.

Second, although prior studies have been conducted to expand understanding of tourist satisfaction, they largely ignored the mediating effects of place attachment on relations between tourist satisfaction and behavioral intentions. Rather, most researchers focused on the direct link between tourist satisfaction and behavioral intentions and overlooked the possibility that place attachment mediates relations between tourist satisfaction and behavioral intentions [2]. Based on existing research on direct effects between these three variables, tourists would seem to be likely to consider 'place attachment' an important component of the path between tourist satisfaction and behavioral intentions. In the present study, to promote theory development, we investigated the mediating effect of tourist satisfaction on the relation between event quality and behavioral intentions, and that of place attachment on the relation between event quality and behavioral intentions. Naturally, the four conditions proposed by Baron and Kenny's [12] were satisfied to establish the validities of mediating effects.

Third, in a sport and tourism context, the majority of studies have focused on large-scale sporting events such as the Olympic Games, the FIFA (Fédération Internationale de Football Association) World Cup, the European Football Championships, and the Asian Games. However, small-scale recurring sporting events have not been well studied. Wong and Tang [5] found small-scale sporting events receive relatively little attention in sport or event tourism literature and argued that such events could lead to loyalty. Hence, it would appear studies on small-scale recurring sporting events add a new perspective to sport tourism literature.

Accordingly, the current study sought to explore mutual relationships between event quality, tourist satisfaction, place attachment, and behavioral intentions collectively and systematically and to place emphasis on their mediating effects at a small-scale recurring sporting event.

## 2. Theoretical Background, Research Hypotheses, and Model

### 2.1. Information about Gyeongju City

Gyeongju is a city near the southeastern coast of mainland South Korea and in 2017 had a population of 257,903 [13]. The city was the capital of the 1000-year-long Silla dynasty, an ancient kingdom, which later governed the Korean peninsula [14]. Thanks to the rich cultural and historical heritage of this dynasty, Gyeongju is known as 'The museum without walls' and has more tombs, temples, rock carvings, pagodas, cultural artifacts, and Buddhist statuaries than any other location in South Korea [15]. In 2000, UNESCO (The United Nations Educational, Scientific and Cultural Organization) designated The Gyeongju Historic Area, which contains the ruins of temples and palaces,

outdoor pagodas and statuary, reliefs, Buddhist art, and the Yangdong Folk Village (a traditional clan village dating back to the Joseon dynasty (added in 2010)), a World Cultural Heritage Site [16].

Gyeongju is one of the most popular tourist destinations for international and domestic visitors, especially among those interested in the cultural heritage of the Silla dynasty and the architecture of the Joseon Dynasty [17]. In addition, Gyeongju has placed emphasis on sporting events, especially marathon events, which allow the city to promote its rich tangible cultural heritage via through TV broadcasts [18]. The Gyeongju International Marathon is held annually and attracts around 20,000 spectators and participants [19], and local authorities are hopeful that those visiting the event will recommend it to others and revisit, and thus, promote Gyeongju as a sporting destination [18].

### 2.2. Event Quality

Undoubtedly, event quality is a crucial topic in sport tourism research and service quality is one of its most important components [11]. Service quality has received much attention in recent years and is recognized as a critical factor because of the influence it has on the psychological and behavioral responses of tourists [20]. Service quality is defined as "a customer's overall impression of the relative inferiority/superiority of an organization and its services" [21]. Based on existing literature, the current study defines event quality as spectator/participant overall judgement regarding the value of an event attribute. In the context of sport tourism, some studies have defined four dimensions of event quality, that is, game quality, interaction quality, outcome quality, and physical environment quality [11]. According to Jin et al. [11], game quality represents spectators' impressions about the quality of the game, interaction quality 'the performance of managing staff and volunteers responsible for delivering services,' outcome quality "the link between consumers' perceptions of gains received from interactions with the service provider," and physical environment quality is the spectators' evaluations of the qualities of physical facilities. Therefore, in the current study, event quality consisted of 12 items representing game quality, interaction quality, outcome quality, and physical environment quality.

### 2.3. Tourist Satisfaction

Tourist satisfaction has been widely shown to be one of the most substantial factors of future success, and much has been published on the topic. The theory of tourist satisfaction is rooted in the marketing literature and according to Oliver [22], "satisfaction is defined as pleasurable fulfillment, that is, the consumer senses that consumption fulfilled some need, desire, goal, or so forth, and that this fulfillment is pleasurable." Some marketing researchers proposed the 'expectancy disconfirmation model', which compares initial expectations and perceived performance after consumption and determines the final state of satisfaction [23]. In other words, if a customer perceives he/she has received more value than expected, he/she is satisfied. Some have suggested the cognitive-affective model, which is regarded as being more insightful because consumer perceived satisfaction is more likely to arise spontaneously by cognitive and affective processing [24]. According to Yoon and Uysal [25], tourist satisfaction can be measured using multiple items, and based on previous studies, Lee [26] proposed three travel satisfaction items—overall satisfaction, satisfaction versus expectation, and satisfaction—based on considerations of invested time and effort. The present study used this scale to measure tourist satisfaction.

A vast amount of the marketing literature is dedicated to the direct relationship between service quality and customer satisfaction [27], and in the context of tourism, some studies have shown that service quality and satisfaction are related [28]. Existing research suggests that if tourists highly value a product or service provided at a sporting event, they are more likely to have a high level of satisfaction. Thrane [29] examined possible links between service quality (festival quality), overall satisfaction, and intention to recommend and concluded service quality is a predictor of satisfaction. Yoon, Lee, and Lee [30] also empirically tested relationships between festival quality, visitor satisfaction, and loyalty using a structural approach and demonstrated that improving festival quality can become a

fundamental strategic metric for building tourist satisfaction. Their findings suggested event quality is related to tourist satisfaction. Accordingly, the current study proposed the following hypothesis.

**H1:** *Event quality influences tourist satisfaction.*

### 2.4. Place Attachment

Place attachment has received much attention in tourism, environmental psychology, and environmental education literature [31]. The word attachment represents affect and the word place represents the environmental setting to which people have strong attachments [9]. According to Morgan [32], place attachment may be defined as "an affective bond to a particular geographical area and to the meaning attributed to that bond." In other words, place attachment is commonly conceptualized as an overall connectedness or bond between a person and a location. In many studies, place attachment has been dichotomized into place identity and place dependence. Place identity refers to "the symbolic importance of a place as a repository for emotions and relationships that give meaning and purpose to life" [33], and place dependence reflects "how well a setting serves goal achievement given an existing range of alternatives" [34].

In the context of hospitality management, empirical evidence shows that event quality is influenced by place attachment. Alexandris, Kouthouris, and Meligdis [35] tested the effect of service quality (physical environment quality, interaction quality, and outcome quality) on place attachment on 264 recreational skiers and showed that physical environment quality and interaction quality play central roles in engendering place attachment. Baek, Ryu, and Chae [36] held the view event quality leads to place attachment. They examined the impact of festival quality on visitor place attachment and behavioral intention and concluded that festival quality is critical for stimulating place attachment. Given the direct impact reported in some studies, it was expected that event quality would positively impact place attachment.

**H2:** *Event quality influences place attachment.*

In line with the development theory of place attachment [32], Hosany et al. [33] acknowledged a link between tourist satisfaction and place attachment. They developed a model based on the development theory of place attachment that provided some useful evidence on the positive effect of tourist satisfaction on place attachment. Su, Cheng, and Huang [37] empirically demonstrated tourist satisfaction is related to place identity and place dependence, and Zenker and Rütter [38] emphasized that satisfaction with a city is a direct antecedent of attachment to the city. Considering previous studies, it seems reasonable to assume that tourist satisfaction positively influences place attachment.

**H3:** *Tourist satisfaction influences place attachment.*

### 2.5. Behavioral Intention

Behavioral intention has long constituted an important domain of research in interdisciplinary studies. The concept can be depicted as a tourists' intention to revisit based on past memorable experiences at a destination and to engage in word-of-mouth [39]. Most researchers have long regarded behavioral intention as one of the most reliable sources of information regarding potential tourists [40]. Lee and Han [41] ascertained behavioral intention can decrease negative recognition and perceived risk of destination, which indicates behavioral intention should be considered when evaluating the success of an event.

The link between event quality and behavioral intention might be explained by a study conducted by Jin et al. [11]. They proposed a conceptual model that included event quality, perceived value, destination image, and behavioral intention, and concluded event quality leads to behavioral intention. Likewise, Moon, Ko, Connaughton, and Lee [42] examined theoretical relationships between service quality at a sporting event, perceived value, destination image, and behavioral intention, and revealed

that service quality significantly influenced behavioral intention. The results of these previous studies hint at a relationship between event quality and behavioral intention, and thus, the current study felt confident enough to postulate a relationship between event quality and behavioral intention.

**H4:** *Event quality influences behavioral intentions.*

The symbolic relationship between satisfaction and behavioral intentions is a frequent topic in interdisciplinary studies [43], and it is generally believed tourist satisfaction leads to behavioral intention. Chen et al. [2] provided supportive evidence for this link and examined relationships between experience quality, perceived value, satisfaction, and behavioral intention for heritage tourists. Their findings revealed that the more satisfied tourists are with experiences, the more likely they are to revisit and recommend the destination to others. Hutchinson, Lai, and Wang [44] reinforced the link between tourist satisfaction and behavioral intentions. An integrative model that explored relationships between quality, value, equity, satisfaction, and behavioral intentions was developed and tested using golf supporters, and it was concluded tourist satisfaction is a strong predictor of behavioral intentions. Therefore, the current study adopted the following hypothesis regarding the impact of tourist satisfaction on behavioral intentions.

**H5:** *Tourist satisfaction influences behavioral intentions.*

The relationship between place attachment and behavioral intention has received renewed attention during the past few years. For example, a tourism study by Lee et al. [9] presented evidence showing that place attachment is related to behavioral intention. They examined the relations between place attachment (place identity and place dependence), festival satisfaction, and loyalty (word-of-mouth, revisit intentions, and destination preference), and detected a significant, positive relationship between place identity and revisit intentions, and between place dependence and word-of-mouth. Loureiro [45] explored the relationship between experience economy, place attachment, and behavioral intentions in the context of rural tourism and maintained place attachment acts as an antecedent of behavioral intentions. Accordingly, it seems reasonable to suggest place attachment affects behavioral intention.

**H6:** *Place attachment influences behavioral intentions.*

With respect to the mediating effect of tourist satisfaction between event quality and behavioral intentions, Lee, Graefe, and Burns [7] demonstrated that satisfaction plays a mediating role between service quality and behavioral intentions. In terms of the mediating effect of place attachment between event quality and behavioral intentions, Alexandris, Kouthouris, and Meligdis [38] indicated that service quality is a key driver of place attachment, which in turn affects consumers' loyalty like behavioral intentions. Also, Jin et al.'s [11] study showed that event quality has a direct effect on behavioral intentions. Regarding the mediating effect of place attachment between tourist satisfaction and behavioral intentions, according to Lee, et al. [9] study, satisfaction indirectly affects word-of-mouth (WOM) via place dependence, and revisit intention via place identity/social bonding of place attachment. Also, existing studies reported satisfaction is a direct antecedent of behavioral intentions [44]. These results strongly support the intervening role of place attachment. Accordingly, the current study proposes the following three hypotheses:

**H7:** *Tourist satisfaction mediates the relationship between event quality and behavioral intentions.*

**H8:** *Place attachment mediates the relationship between event quality and behavioral intentions.*

**H9:** *Place attachment mediates the relationship between tourist satisfaction and behavioral intentions.*

Based on previous studies, the current study proposes the following conceptual model (Figure 1).

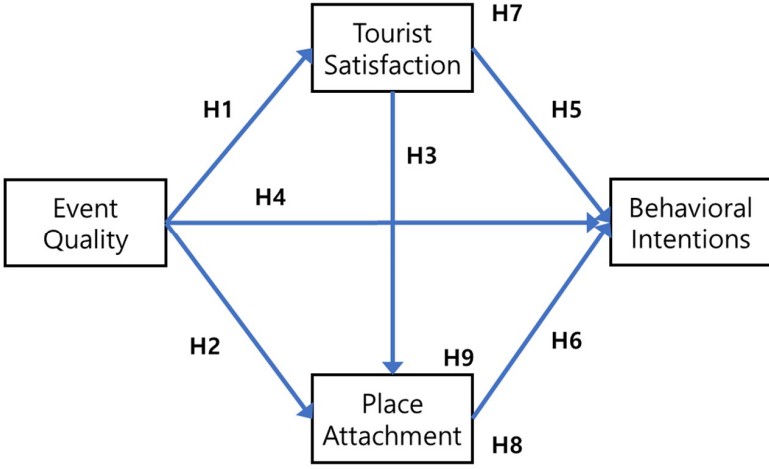

**Figure 1.** The conceptual model.

*2.6. What is Sport Tourism and How It Divides?*

Sport tourism has resurged in the past few years as it makes an important contribution to local economy such as improving destination image and attracting a flow of visitors [46]. In fact, many local governments in South Korea view sport tourism as one of the most critical driving forces of tourism success, and strive to host sport events. According to Gibson's [47] study, sport tourism can be described as leisure-based travel that takes individuals temporarily away from their mundane life to participate in or watch a sporting event or to venerate a sporting heritage. Therefore, the author suggests that sport tourism can be described as three main behaviors: participating, watching, and visiting/venerating. Based on this view, there is explicit consensus among sport tourism researchers that sport tourism is divided into three categories: event sport tourism, active sport tourism, and nostalgia sport tourism [47]. Event sport tourism depicts tourists passively engage in a sporting event as spectators whose main purpose of travel is to cheer for their country and favorite star players or to spend time with family members, friends, and social groups [48]. As such, many countries and regions increasingly consider mega sporting events such as Olympic Games and World Championships as a core component of the destination marketing strategy [6]. Active sport tourism describes tourists actively engage in a sporting event as participants in order to improve mental and physical health, relieve the stress, and meet personal goals [49] (Figure 2). Nostalgia sport tourism represents tourists visit famous sporting venues such as stadiums and museums of the mega sporting events and professional teams [47].

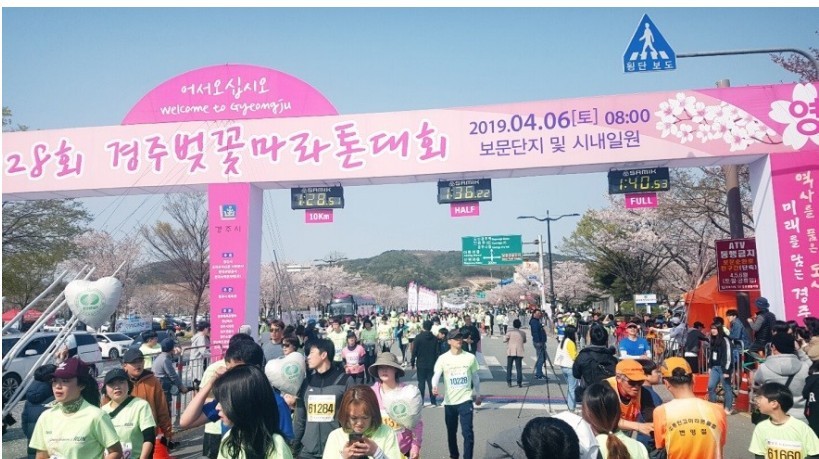

**Figure 2.** Active sport tourism.

## 3. Method

### 3.1. Data Collection

Data for the current study was collected from spectators and participants that attended the 2018 International Marathon in Gyeonju City. To collect a representative sample, the authors and two trained research assistants administered a face-to-face, questionnaire-based survey on 21 October 2018. Using a convenient sampling procedure, 400 people were approached to take part in in the survey. No-one was solicited to participate. Finally, 375 questionnaires were completed, but 25 were subsequently eliminated due nonanswered questions. The remaining 350 satisfactorily completed questionnaires were analyzed. The sample was almost equally split between makes (64%) and females (36%). The majority were aged between 40 and 49 years (40.3%, *n* = 141), university educated (44.3%, *n* = 155), married (68.6%, *n* = 240), and earned $40,000–59,999 (21.7%, *n* = 76) per annum.

### 3.2. Measures

The survey instrument was developed based on the study objectives and a literature review. A seven-point Likert scale, anchored on unimportant and extremely important (scored as 1 and 7, respectively) was used. The survey questionnaire addressed five domains: (a) event quality, (b) tourist satisfaction, (c) place attachment, (d) behavioral intention, and (e) demographic information. Event quality was assessed using 12 items (3 items addressed game quality, 3 interaction quality, 3 outcome quality, and 3 physical environment quality); these items were adapted from those used by Jin et al. [11] and by Jae Ko, Zhang, Cattani, and Pastore [50]. Tourist satisfaction was assessed using 3 items derived from Yoon et al. [25]. Place attachment was assessed using 6 items (3 items addressed place identity and 3 place dependence), and these items were adapted from Hosany et al. [33] and Williams and Vaske [51]. Behavioral intention was assessed using 6 items (3 items addressed intention to recommend and 3 intention to revisit), and these items were adapted from Hosany et al. [33], Žabkar, Brenčič and Dmitrović [52], and Lam and Hsu [53]. To ensure face validity, three experts with theoretical knowledge were invited to examine the survey items.

### 3.3. Validity and Reliability

The nine-factor (game quality, interaction quality, outcome quality, physical environment quality, tourist satisfaction, place identity, place dependence, and intention to recommend and intention to revisit) confirmatory factor analysis (CFA) model used had a total of 288 degrees of freedom. The measurement model indicated reasonable fit with data ($x^2/df$=2.531, NFI = 0.923, RFI = 0.907, TLI = 0.941, SRMR = 0.046 and RMSEA = 0.066). All model fit indices were within recommended thresholds [54] (Table 1).

Construct validity was assessed using convergent and discriminant validities. To evaluate convergent validity, we calculated factor loadings, construct reliability (CR), and average variance extracted (AVE). All factor loading values were < 0.5 and significant ($p < 0.001$) (Table 1). CR values all exceeded the recommended value of 0.7 (range from 0.827 to 0.914) and AVE values all exceeded the minimum requirement of 0.5 (range 0.615 to 0.780). Therefore, convergent validity was satisfactory (Table 1).

If all the items in the structural equation model are used as observed variables, the complexity of the model increase, which may cause problems such as the size of the sample, the model fit indices, and the significance of the parameter estimation [55]. When there are too many items, the number of items should be adjusted through item parceling [56]. Item parceling is a method of parceling using average when analysis is difficult in the structural equation model due to the large number of observable variables and has become increasingly popular in various area such as education, psychology, and marketing in recent years [56]. To utilize item parceling, convergent validities regarding the three observable variables of game quality, interaction quality, outcome quality, physical environment

quality, place identity, place dependence, intention to recommend, and intention to revisit should be satisfactory.

**Table 1.** Confirmatory factor analysis (CFA) results for measurement model.

| Scale Items | $\beta$ | CR | AVE | Cronbach |
|:---|:---:|:---:|:---:|:---:|
| **Game quality** | | | | |
| It is exciting to watch skillful players | 0.736 | | | |
| Skill performance of players is excellent | 0.763 | 0.827 | 0.615 | 0.783 |
| Information about this evet is easy to obtain | 0.750 | | | |
| **Interaction quality** | | | | |
| The demeanor of the staff is pleasant | 0.841 | | | |
| I enjoy being with the other spectators | 0.888 | 0.862 | 0.676 | 0.860 |
| Spectators follow the regulations | 0.733 | | | |
| **Outcome quality** | | | | |
| I view the outcome of this event favorably | 0.864 | | | |
| I really enjoy the social interaction at this event | 0.831 | 0.863 | 0.677 | 0.895 |
| I spend quality time with my friend/family at this event | 0.884 | | | |
| **Physical environment quality** | | | | |
| The facility is clean and well maintained | 0.852 | | | |
| I am impressed with the design of the facility | 0.891 | 0.880 | 0.709 | 0.904 |
| The facility is safe | 0.876 | | | |
| **Tourist satisfaction** | | | | |
| Gyeongju is better than I expected | 0.866 | | | |
| It is worth visiting Gyeongju | 0.914 | 0.870 | 0.691 | 0.918 |
| Overall, I am satisfied with holidaying in Gyeongju | 0.893 | | | |
| **Place identity** | | | | |
| Gyeongju is a very special destination to me | 0.894 | | | |
| I feel very attached to Gyeongju | 0.909 | 0.870 | 0.691 | 0.910 |
| Holidaying in Gyeongju means a lot to me | 0.845 | | | |
| **Place dependence** | | | | |
| GJ is the best for what I like to do on holidays | 0.816 | | | |
| I would not substitute GJ with any other places | 0.884 | 0.849 | 0.654 | 0.784 |
| I got more satisfaction out of holidaying in GJ than others | 0.736 | | | |
| **Intention to recommend** | | | | |
| I will recommend Gyeongju to other people | 0.930 | | | |
| I will say positive things about Gyeongju to other people | 0.960 | 0.903 | 0.756 | 0.955 |
| I will encourage friends and relatives to visit Gyeongju | 0.920 | | | |
| **Intention to revisit** | | | | |
| If had to decide again I would choose GJ again | 0.970 | | | |
| I want to visit Gyeongju | 0.922 | 0.914 | 0.780 | 0.956 |
| I intend to visit Gyeongju in next 12 months | 0.918 | | | |
| $x^2/df$ = 2.531, NFI = 0.923, RFI = 0.907, TLI = 0.941, SRMR = 0.046 and RMSEA = 0.066 | | | | |

To evaluate convergent validities, we confirmed factor loadings, CR and AVE whose values all exceeded the recommended values (Table 1). Since the convergent validities concerning all observable variables was satisfactory, these variables were parceled as event quality, place attachment, and behavioral intentions, respectively. In other word, each subfactor of event quality, place attachment, and behavioral intentions turned into four, two, and two observable variables, respectively.

To evaluate discriminant validity, we compared the square root of AVE for each construct with correlations between pairs of latent variables [57]. Since it was difficult to verify all latent variables, the pair with the highest correlation was selected and verified. The highest correlation found was 0.729 (between tourist satisfaction and behavioral intentions) (See Table 2) and its square value $(0.729)^2$ was 0.531. AVE for tourist satisfaction was 0.692 and AVE for behavioral intentions was similar at 0.670. Since AVE values were all greater than the square value (0.531), discriminant validity was satisfactory.

Reliability estimates (Cronbach's alpha) for game quality, interaction quality, outcome quality, physical environment quality, tourist satisfaction, place identity, place dependence, intention to recommend, and intention to revisit were above the recommended threshold of 0.7 (range from 0.783 to 0.956), indicating measures were reliable (Table 1).

**Table 2.** Correlations between constructs.

| Constructs | 1 | 2 | 3 | 4 |
|---|---|---|---|---|
| Event quality | 1 | | | |
| Tourist satisfaction | 0.686 ** | 1 | | |
| Place attachment | 0.415 ** | 0.596 ** | 1 | |
| Behavioral intentions | 0.606 ** | 0.729 ** | 0.621 ** | 1 |

** $p < 0.01$.

## 4. Results

Overall, the structural equation modeling (SEM) achieved acceptable fit ($x^2 = 0.103.093$, $df = 38$, $x^2/df = 2.713$, $p < 0.001$). The absolute fit index (Goodness of Fit Index = 0.951, Standardized Root Mean Residual = 0.031, Adjusted Goodness of Fix Index = 0.915 and Root Mean Error of Approximation = 0.07), and incremental fit index (Normed Fit Index = 0.968 and Relative Fit Index = 0.954) were satisfactory.

**Table 3.** Structural parameter estimates.

| Hypothesis | Path | Coefficient | *t*-value | Results |
|---|---|---|---|---|
| 1 | Event quality → tourist satisfaction | 0.736 | 13.775 *** | Accepted |
| 2 | Event quality → place attachment | 0.112 | 2.063 * | Accepted |
| 3 | Tourist satisfaction → place attachment | 0.647 | 11.224 *** | Accepted |
| 4 | Event quality → behavioral intentions | 0.143 | 2.690 ** | Accepted |
| 5 | Tourist satisfaction → behavioral intentions | 0.436 | 5.461 *** | Accepted |
| 6 | Place attachment → behavioral intentions | 0.420 | 5.150 *** | Accepted |

* $p < 0.05$, ** $p < 0.01$, *** $p < 0.001$.

The estimates of structural coefficients (paths) provided the basis for testing the proposed hypotheses. As shown in Table 3, event quality had a significantly positive effect on tourist satisfaction (0.736, $p < 0.001$) and place attachment (0.112, $p < 0.05$), which offers supportive evidence for hypotheses 1 and 2. The path coefficient from tourist satisfaction to place attachment was positive and statistically significant (0.647, $p < 0.001$), supporting hypothesis 3. Hypothesis 4 was also supported, as event quality significantly and positively influenced behavioral intentions (0.143, $p < 0.01$). Tourist satisfaction and place attachment were found to influence behavioral intentions directly (0.436 and 0.420, respectively, $p < 0.001$), which supported hypotheses 5 and 6.

As shown in Table 4, the mediating effect of tourist satisfaction on the relation between event quality and behavioral intentions was found to be statistically significant. In other words, tourist satisfaction showed a partial mediating effect, which supported hypothesis 7. Furthermore, the mediating effect of place attachment on the relation between event quality and behavioral intentions was found to be statistically significant. In other words, place attachment showed a partial mediating

effect, which supported hypothesis 8. The mediating effect of place attachment on the relation between tourist satisfaction and behavioral intentions was found to be statistically significant, that is, place attachment showed a partial mediating effect, which supported hypothesis 9.

**Table 4.** Mediating effects of tourist satisfaction and place attachment.

| Path | | Coefficient | S.E. | 95% CI (Bias-corrected) | p |
|------|------|------|------|------|------|
| Event quality → Tourist satisfaction → Behavioral intentions | Indirect effect | 0.406 | 0.042 | 0.328 ~ 0.468 | 0.004 |
| | Direct effect | 0.200 | 0.069 | 0.093 ~ 0.308 | 0.006 |
| | Total effect | 0.606 | 0.052 | 0.517 ~ 0.693 | 0.003 |
| Event quality → Place attachment → Behavioral intentions | Indirect effect | 0.185 | 0.113 | 0.056 ~ 0.392 | 0.008 |
| | Direct effect | 0.421 | 0.125 | 0.213 ~ 0.598 | 0.001 |
| | Total effect | 0.606 | 0.052 | 0.517 ~ 0.693 | 0.003 |
| Tourist satisfaction → Place attachment → Behavioral intentions | Indirect effect | 0.172 | 0.129 | 0.047 ~ 0.413 | 0.011 |
| | Direct effect | 0.557 | 0.139 | 0.304 ~ 0.712 | 0.001 |
| | Total effect | 0.729 | 0.032 | 0.673 ~ 0.776 | 0.003 |

As shown in Table 4, the mediating effect of tourist satisfaction on the relation between event quality and behavioral intentions was found to be statistically significant. In other words, tourist satisfaction showed a partial mediating effect, which supported hypothesis 7. Furthermore, the mediating effect of place attachment on the relation between event quality and behavioral intentions was found to be statistically significant. In other words, place attachment showed a partial mediating effect, which supported hypothesis 8. The mediating effect of place attachment on the relation between tourist satisfaction and behavioral intentions was found to be statistically significant, that is, place attachment showed a partial mediating effect, which supported hypothesis 9.

## 5. Discussion and Conclusions

The main objective of this study was to investigate integrated model positive relationships between event quality, tourist satisfaction, place attachment, and behavioral intentions with emphasis on the mediating effects of tourist satisfaction and place attachment at a small-scale recurring sporting event. The proposed model allows the identification of relationships between (1) event quality and tourist satisfaction, (2) event quality and place attachment, (3) tourist satisfaction and place attachment, (4) event quality and behavioral intentions, (5) tourist satisfaction and behavioral intentions, and (6) place attachment and behavioral intentions, and showed (7) tourist satisfaction and place attachment had partial mediating effects on the relation between event quality and behavioral intentions, and that (8) place attachment partial mediated the relation between tourist satisfaction and behavioral intentions.

From a theoretical point of view, this study provides several contributions to research in marketing, tourism, hospitality, and sport management. First, this study is a response to recent calls to researchers to consider event quality as an antecedent that influences outcome variables such as tourist satisfaction, place attachment, and behavioral [11,42]. In other words, our findings demonstrate the merit of including event quality in studies aimed at better understanding and predicting tourist behaviors, whereas previous studies have almost exclusively considered destination image and tourist satisfaction as determinants of behavioral intentions [40,58]. Furthermore, no previous study has explored structural relationships between event quality, tourist satisfaction, place attachment, and behavioral intentions. Furthermore, our study delineates four dimensions (game quality, interaction quality, outcome quality, and physical environment quality) and suggests 12 items that can be used to accurately reflect the characteristics of sporting event quality [11].

Second, this study sheds new light on the link between tourist satisfaction and place attachment. In fact, the relationship between tourist satisfaction and place attachment is the subject of debate. Some researchers insist satisfaction be conceptualized as antecedent of place attachment [59], whereas

others argue that satisfaction be conceptualized as an outcome of place attachment [60]. Based on a comprehensive analysis by Morgan [32] on place attachment, Hosany et al. [33] held the view that tourist satisfaction is an antecedent of place attachment. Likewise, Lee et al. [9] found "to a greater extent, satisfied visitors become attached to a community due to attributes and features that support their experience-related needs." Thus, the consensus appears to be that tourist satisfaction is an antecedent of place attachment.

Third, this study heeds the call of Loureiro [45] by seeking to understand the role of place attachment in a tourist behavioral model. More specifically, Loureiro emphasized place attachment is central to the understanding of behavioral intentions and urged further investigation be conducted on the relationship. In this respect, the present study creates a link between place attachment and behavioral intentions by showing that an affective bond with the destination, based on past meaningful and memorable experiences, may lead to an intention to engage in word-of-mouth communications and revisit. Also, we tested the relationship between place attachment and behavioral intentions, and found place attachment is an important predictor of behavioral intentions among sport tourists, which is in accord with previous studies [9,61].

Fourth, this study contributes to tourism studies by examining the mediating effects of tourist satisfaction and place attachment on the relation between event quality and behavioral intentions. Our results highlight the need to include event quality, tourist satisfaction, and place attachment in models aimed at determining sport tourists' intentions to recommend and revisit a destination. Notably, our findings demonstrate tourist satisfaction partially mediates place attachment. The relations found in the present study make an important contribution to the sport tourism literature because they reveal that while event quality plays a pivotal role in nurturing behavioral intentions, tourist satisfaction and place attachment also play key roles by strengthening behavioral intentions. Furthermore, they indicate intention to recommend and revisit can be developed through tourist satisfaction and place attachment as well as by event quality.

Fifth, the study also revealed place attachment mediated the relationship between tourist satisfaction and behavioral intentions. In particular, although the proposed model was grounded in extant literature, the present study is the first to incorporate place attachment as a mediator of the relation between satisfaction and behavioral intentions. Furthermore, the results obtained show place attachment has a partial mediating effect and confirm that tourist satisfaction indirectly influences behavioral intentions via place attachment. Previous studies have overlooked the mediating role of place attachment on the path from tourist satisfaction to behavioral intentions. By exploring the mediating effect of place attachment, the present study helps to address a gap in current tourism marketing literature and represents an important step toward understanding tourist psychology and behavior.

From a practical point view, the results of the present study provide meaningful guidelines to destination marketers in the context of sporting tourism destinations. First of all, they demonstrate that strengthening the quality of an event should be considered a priority by destination marketers. Our findings demonstrate that event quality has a positive effect on tourist satisfaction, place attachment, and behavioral intentions, and should encourage destination marketers to make strenuous efforts to improve the four latent dimensions of event quality (i.e., game quality, interaction quality, outcome quality, and physical environment quality) to meet the needs and desires of sporting tourists.

To improve game quality, event organizers should endeavor to attract talented and competent players to participate. It is intuitively assumed that if competent players are involved, tourists' interests in the event will be increased. To enhance interaction quality, volunteers and staff should be trained and educated well, because acts of kindness can have considerable positive impacts on tourists [33], whereas displays of a negative demeanor or inhospitality result in negative attitudes toward the destination and event. To develop physical environment quality, destination managers should increase safety-orientated security and maintain the cleanliness of facilities. Constant efforts are required to develop positive perceptions of environment quality and to ensure the success of

sporting events. In addition, the current study suggests hosting events offer excellent educational opportunities for vocational college and university students, especially those majoring in marketing, tourism, hospitality, or sport management. Students could participate as volunteers to help and communicate with visitors. Such efforts would contribute to the development of event quality and help create positive behavioral intentions.

Several limitations of this study warrant consideration. First, the cohort was composed individuals visiting the venue for the first-time and repeat visitors, which may differ in terms of travel characteristics, travel motivations, and post-trip evaluations [33]. The current study suggests this aspect be explored in the future research by multigroup analysis. Second, the proposed model includes only a handful of constructs, and thus, to more fully understand sport tourist psychology and behaviors, we suggest additional antecedents (e.g., emotional experience, destination image, and motivation) of tourist satisfaction and behavioral intentions be included in future studies. Third, relations between tourist satisfaction and place attachment were investigated to determine the path between event quality and behavioral intentions, and there is a possibility that other variables such as perceived value and brand image also have mediating effects. These variables should also be addressed to provide a more comprehensive framework.

**Author Contributions:** Conceptualization, Y.J. and S.K.; Methodology, Y.J. and J.Y.; Software, Y.J. and J.Y.; Validation, J.Y. and S.K.; Formal Analysis, Y.J.; Investigation, S.K.; Resources, Y.J.; Data Curation, S.K.; Writing—Original Draft Preparation, Y.J.; Writing—Review and Editing, Y.J. and S.K.; Visualization, J.Y.; Supervision, J.Y.; Project Administration, S.K.; Funding Acquisition, J.Y.

**Funding:** This research received no external funding.

**Conflicts of Interest:** The authors declare no conflicts of interest.

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
