# Peer review of "Determinants of Behavioral Intentions in the Context of Sport Tourism with the Aim of Sustaining Sporting Destinations"

_sustainability, doi:10.3390/su11113073_

Round 1
Reviewer 1 Report
Interesting article regarding targeted marketing of average size city in Korea. The authors assumed that sporting events are available at attractive prices in Gyeongju City. The relationshops between the quality of events, tourists satisfacion and their behaviors were examined. Admittedly, the authors have stated that they are only interested in small sports events, but it is worth quoting in the literature review some of the research regarding big sports events such as Olympic Games or World Championships. In addidtion, sport events are included in sports tourism theory. In my opinion, there was no mentioning what sports tourism is and how it divides.
In my opinion, sports tourism can be treated as a separate type, like culinary, film or other forms of cultural tourism as broadly understood. It has already been described by different authors attempting to define and study it. It encompasses all forms of active and passive engagement in sports activities performed away from the place of residence and work (STANDEVEN & DE KNOP 1999). Participating in sports tourism passively or actively depends on the main motivations to choose this type of travel. Depending on whether they are sport or tourism focused (GAMMON & ROBINSON 2004), they divide participants into those who undertake sports tourism with the stress placed on either sport or tourism. In the first case, the participants are more strongly motivated by sport and in the other by tourism.
Sports tourism may become the driving force behind social and economic development. Sporting events and active sports tourism may diversify the tourism product and increase its innovativeness, and as a result – the tourists’ satisfaction, the profits from economic activity and the share in the market. They may also create a new image and increase the attractiveness of the tourist offer. Considering the above, I forecast further, dynamic growth in sports tourism.
The reviewed article concerns passive sports tourism, fans of marathon runs and their behaviors as well as the preferences of tourists visiting the city of Gyeongju.
Gammon S., Robinson T., 1997, Sport and Tourism: A conceptual Framework, „Journal of Sport Tourism”, Oxford, 4, (3), s. 21-26
Standeven J., DeKnop, P., 1999, Sport Tourism, Human Kinetics, Champaign.
Ross D., 2001, Developing Sports Tourism. An eGuide for Destination Marketers and sports Events Planners, University of Illinois: National Laboratory for Tourism and eCommerce.
Author Response
Thank you for us the opportunity to revise and resubmit our manuscript. Following your suggestions, we have added some information about what sport tourism is and how it divides. We hope you find this revision satisfactory.

Reviewer 2 Report
The study described in this paper has an interesting and current theme in the context of scientific research in Sports Tourism. However, there are a number of aspects in it that need to be improved and which I suggest next.
The authors develop a series of hypotheses in part 2 of the paper. However, hypotheses H7 to H9 are not duly justified. Each of them should be justified according to the literature. However, reading Part 3, Method, and 4. Results reveal that there is no coherence between the content of all of them.
The hypotheses are developed and justified in a generic way. In the conceptual model event quality synthesizes 4 constructs - game quality, interaction quality, physical environment quality -, place attachment synthesizes 2 - place identity and place dependence -, and behavioral intentions summarizes - intention to recommend and intention to revisit - as can be observed in table 1.
However, Table 1 and the CFA model, base model of all other analyses performed, is developed with 9 constructs, the remaining results presented synthetically in Table 2 to 4 are performed with only 5 constructs.
Instead, authors should maintain consistency across all different parts of the paper and develop the entire analysis, including the hypothesis, based on the 9 constructs.
In this way, I cannot confirm the results and conclusions obtained.
Minor Things
- Do not use "we" as a personal pronoun. Instead, try using a more impersonal way.
- Change the number of the "results" part from 3 to 4, as well as the "discussion and conclusions" from 4 to 5.
- In table 2 use the word constructs instead of variables.
Author Response
Thank you for us the opportunity to revise and resubmit our manuscript. We found your comments to be insightful and helpful. We hope you find this revision satisfactory.

Reviewer 3 Report
The study is correct, the methodology is well explained and the results and conclusions are coherent. x2 / fd is close to 3, but it is acceptable.
Author Response
Thank you very much!
Round 2
Reviewer 2 Report
I think the authors did not sufficiently justify the hypotheses H7 to H9 in a sufficiently and individualised way. For example, what it means "In terms of the mediating effect of place attachment between event quality and behavioral intentions, some studies show that service quality could indirectly influence willingness to revisit and recommend." What are these studies?
Regarding the other aspects, I keep all my previous comments. I do not understand how the authors introduced in the paper the "Item parceling" method to which the comments refer. I also do not find in the paper any relation between this method and that developed by the authors. I think the authors did not respond to the different aspects I mentioned in the first review of the paper.
Author Response
Thank you for us the opportunity to revise and resubmit our manuscript. Follwoing your suggestions, we have revised the hypotheses H7 to H9 and added information about item parceling. We truly hope you find this revision satisfactory.
